# Subthreshold Conduction of Disordered ZnO-Based Thin-Film Transistors

**DOI:** 10.3390/mi14081596

**Published:** 2023-08-13

**Authors:** Minho Yoon

**Affiliations:** Department of Physics and Institute of Quantum Convergence Technology, Kangwon National University, Chuncheon 24341, Republic of Korea; minhoyoon78@gmail.com

**Keywords:** subthreshold swing, localized trap states, ZnO, thin-film transistors

## Abstract

This study presents the disorderedness effects on the subthreshold characteristics of atomically deposited ZnO thin-film transistors (TFTs). Bottom-gate ZnO TFTs show n-type enhancement-mode transfer characteristics but a gate-voltage-dependent, degradable subthreshold swing. The charge-transport characteristics of the disordered semiconductor TFTs are severely affected by the localized trap states. Thus, we posit that the disorderedness factors, which are the interface trap capacitance and the diffusion coefficient of electrons, would result in the degradation. Considering the factors as gate-dependent power laws, we derive the subthreshold current–voltage relationship for disordered semiconductors. Notably, the gate-dependent disorderedness parameters are successfully deduced and consistent with those obtained by the *g_m_*/*I_ds_* method, which was for the FinFETs. In addition, temperature-dependent current–voltage analyses reveal that the gate-dependent interface traps limit the subthreshold conduction, leading to the diffusion current. Thus, we conclude that the disorderedness factors of the ZnO films lead to the indefinable subthreshold swing of the ZnO TFTs.

## 1. Introduction

In recent decades, oxide semiconductors such as zinc oxide (ZnO), indium–gallium–zinc oxide (IGZO), and indium–zinc oxide (IZO) have received considerable attention due to their massive potential for applications in flexible displays, logic circuits, and wearable devices [1,2,3,4]. Especially the subthreshold voltage operation of oxide-based TFTs has been an emerging technology for low-power applications [5]. Hence, efforts to achieve steep-subthreshold-swing oxide TFTs have led to several proposed approaches, such as applying Schottky source/drain contacts [6], implementing a high-*K* dielectric such as HfO_2_ [7], and depositing an in situ Al_2_O_3_ gate insulator [8]. Although there has been progress, the observed steep subthreshold swings were generally minimally extracted values, thereby rendering their application to the devices. Moreover, probably due to the localized states in disordered oxide semiconductors [9,10,11], the experimental and theoretical characterization of the subthreshold conduction characteristics of the disordered semiconductors has not been well established compared to those of metal-oxide-semiconductor field-effect transistors (MOSFETs) [12,13,14]. Additionally, capacitance–voltage (*C–V*) based methods such as the low- and high-frequency methods [15,16], the conductance method [17], and the charge pumping method [18] could be used for characterizing the subthreshold characteristics of the disordered semiconductors. However, it should be carefully applied since the deposition of electrodes such as aluminum can cause electron doping on the oxide semiconductors, which alters the subthreshold characteristics [19,20,21,22]. Furthermore, due to the low capacitance of the generally used Si/SiO_2_ wafers (<~20 n Fcm^−2^), the capacitance changes from the interface or semiconductor traps would be hardly probed. Hence, for these reasons, current–voltage-based analyzing methods are believed to be strongly required for disordered oxide semiconductors. Please note that the widely used relation for characterizing the interface traps from the subthreshold swing, *S*.*S* = ln10*kT*/*q*(1 + *q*^2^*D_it_*/*C_i_*), was developed regarding the interface traps as constant over the energy states [23].

In this study, we have explored the subthreshold conduction characteristics of ZnO TFTs. Bottom-gate, atomic-layer-deposited ZnO TFTs show typical n-type transfer characteristics [19]. However, the subthreshold swing seems to be seriously degraded: as the gate bias increases, the subthreshold swing increases as extracted to 1.8 V/dec at *V_gs_* = 9 V but to 8.1 V/dec at *V_gs_* = 15 V. We posit that the disorderedness of the ZnO films would result in the degradation; hence, we derived the subthreshold current–voltage relationship by considering that the disorderedness factors, which are the interface trap capacitance and the diffusion coefficient of electrons, can be modeled with gate-dependent power laws [24,25,26]. Following the relationships, the gate-voltage-independent disorderedness factors were successfully extracted and consistent with those obtained by the *g_m_*/*I_ds_* method [27]. In addition, we can identify that the interface traps hinder the subthreshold charge transport of the ZnO films by comparing the interface trap density with the density of states of ZnO TFTs. Therefore, we can state that the disorderedness factors lead to the gate-dependent subthreshold characteristics of the ZnO TFTs and can be extracted in detail using current–voltage-based analyzing methods.

## 2. Materials and Methods

Bottom-gate, top-contact (BGTC) ZnO TFTs were fabricated on a 10 nm thin, Al_2_O_3_-coated, 200 nm thick p^+^-Si/SiO_2_ substrate. A 6 nm thin ZnO film was deposited by cyclic atomic layer deposition (ALD) at 80 °C and patterned by a conventional lift-off photolithographic process. Diethylzinc (DEZ, Aldrich, St. Louis, MO, USA) and water were used as zinc and oxygen precursors, respectively. Then, a 50 nm thick Al source and drain electrodes were deposited by thermal evaporation and patterned using a shadow mask, which had a width and length of 1000 and 360 μm, respectively. To extract the intrinsic conductance of the ZnO channel, a four-point probe source and drain configuration was used, of which the voltage-probing electrodes (*V*_1_ and *V*_2_) were placed at 120 and 240 μm in the channel. The geometric capacitance of the dielectric was measured to be 16.4 nF cm^−2^ at 1 kHz, using an LCR meter (HP4284A, Agilent Technologies, Santa Clara, CA, USA). The thickness of the films was measured with an ellipsometer (AutoEL-II, Rudolph Research, Hackettstown, NJ, USA) and confirmed with an atomic force microscope (XE100, Park Systems, Suwon, Republic of Korea). The current–voltage (*I–V*) characteristics of the transistors were investigated with a semiconductor parameter analyzer (Model HP4155C, Agilent Technologies, Santa Clara, CA, USA). A liquid nitrogen cooling cryostat was used for temperature-variable current–voltage measurements. The temperature range was from 180 to 300 K. The extrinsic field-effect mobility of the TFTs in a linear regime was estimated using the following Equation (1):(1)$$\mu_{lin}=\frac{1}{C_{i}V_{ds}}\frac{L}{W}\frac{\partial I_{ds}}{\partial V_{gs}}$$
where *I_ds_* is the drain current, *V_gs_* is the gate voltage, *C_i_* is the geometric dielectric capacitance, *V_ds_* is the drain voltage, and *L* and *W* are the channel length and width, respectively.

For extracting the potential distributions of the ZnO TFTs, we use the 4-probe method. By considering that the potential in the channel is linearly distributed, the potential drops at the source and drain electrodes (Δ*V_S_*, Δ*V_D_*) can be deduced by measuring potentials with the voltage probes (*V*_1_ and *V*_2_), which are given by Equation (2):(2)$${\Delta V}_{S}=V_{1}-\frac{\left(V_{2}-V_{1}\right)}{\left(L_{2}-L_{1}\right)}\left(L_{1}\right),\;\;\;\;{\Delta V}_{D}=V_{D}-\left[V_{2}+\frac{\left(V_{2}-V_{1}\right)}{\left(L_{2}-L_{1}\right)}\left(L-L_{2}\right)\right]$$
where *V*_1_ and *V*_2_ are measured voltages with potential probes, and *L*_1_, *L*_2_, and *L* are the distance from the source electrode to the first, second, and drain electrodes, respectively. Thus, we extract the potential distributions of the ZnO TFTs using the potential drops and measured potentials. In this study, the voltage-probing electrodes (*V*_1_ and *V*_2_) were placed at 120 and 240 μm in the channel.

Thin-film crystallinity was measured by X-ray diffraction (XRD) using a high-resolution X-ray diffractometer (Smartlab, Rigaku, Akishima-shi, Tokyo, Japan) with a HyPix-3000 detector and Cu-Kα (λ = 1.54 Å) radiation operating at 9 kW. The surface morphologies of the films were scanned with an atomic force microscope (XE100, Park Systems, Suwon, Republic of Korea).

## 3. Results

Figure 1a shows the typical n-type transfer characteristics (*I_ds_* vs. *V_gs_* at *V_ds_* = 1 V) of the atomically deposited ZnO thin-film transistors (TFTs). The field-effect mobility (*μ*) from the transconductance is extracted to be 3.1 cm^2^ V^−1^ s^−1^ at *V_gs_* = 80 V, and a threshold voltage (*V_th_*) is estimated to be 14.9 V along with a high on/off ratio (>10^4^). However, as depicted in Figure 1b, the subthreshold swing (*S*.*S*) of the ZnO TFTs is difficult to be determined. By using the relation of *S*.*S* = *dV_gs_*/*dlogI_ds_* [14,23], the subthreshold swing of the ZnO TFTs is minimally extracted to be 1.8 V/dec at *V_gs_* = 9 V; however, as the gate bias increases, it significantly increases to 5.3 V/dec at *V_gs_* = 12 V, reaching 8.1 V/dec at *V_gs_* = 15 V. As a result, although the minimum value was 1.8 V/dec, the required gate voltage is 2.1 V for the increase from 10^−10^ to 10^−9^ A and 3.8 V for the increase from 10^−9^ to 10^−8^ A. The subthreshold swing is defined as the required gate voltage for changing the drain current by one order of magnitude [14]. In the case of metal-oxide-semiconductor field-effect transistors (MOSFETs), the subthreshold swing exhibits a relatively constant value and is thereby utilized for determining the sharpness of the on-to-off transition [27,28]. However, in the case of the ZnO TFTs, due to the detrimental increment, the minimally deduced value cannot be used for evaluating the sharpness. Please note that the ZnO films exhibit the typical polycrystalline structure, which was mainly oriented to the (002) direction (34.50°) as in Appendix A. For the wurtzite-structure ZnO, the c-axis (002) plane has been known as the most densely packed and thermodynamically stable orientation [29,30]. We conducted the XRD analysis of the ZnO film using the stacked structure as in the inset of Appendix A, because of the penetration depth. In addition, we have investigated the surface morphology of the ZnO films. As seen in the AFM image in Appendix A, the thin-film structure shows the granny microstructure with the root-mean-square roughness of the films of ~0.5 nm. Hence, we regard the structure of the ZnO as a polycrystalline structure. Appendix A presents the corresponding output characteristics of the ZnO TFTs and the thickness of the films confirmed with an atomic force microscope.

The contact resistance of the ZnO TFTs could be one possible reason for the degradation [24,31]. If the contact resistance of the ZnO TFTs is high enough to result in the weighty potential drops at the source and drain electrodes, the drain current could be significantly altered, resulting in degradation. Hence, we investigate the intrinsic subthreshold swing of the ZnO TFTs using the contact-effect decoupled transfer characteristics. Figure 2a shows the potential distributions of the ZnO TFTs, which were extracted by the four-probe method as in the inset [19,32]. Please see the Materials and Methods section for more information on the four-probe method. At the low gate bias (11 V), the potential drop at the source electrode was as high as 0.5 V and gradually decreased to 0.1 V at the threshold voltage (14.9 V). Above the threshold voltage, the potential drops were negligible (<0.1 V). Thus, the effective applied channel potential was deduced to be as low as 0.3 V at the low bias but as high as 0.8 V at the threshold voltage. After normalizing the drain current with the observed effective channel bias, the intrinsic subthreshold swing was retraced in Figure 2b. However, although decoupled, the degradation of the subthreshold can still be observed, which was deduced to be 1.7 V/dec at *V_gs_* = 9 V and 9.4 V/dec at *V_gs_* = 15 V, respectively. To our best knowledge, this would be attributed to the fact that the subthreshold conduction was mainly driven by diffusion [14,23], which can be expressed as follows. By considering the current from the concentration gradient at the source, drain electrodes can be expressed in Equations (3) and (4), and concentrations (*n*) at the electrodes can be estimated by the Boltzmann approximation in Figure 3a, the subthreshold currents can be generally expressed as in Equation (5) [23,33]. When the *V_ds_* is greater than a few *q*/*kt*, the latter term of (1 − exp(−*qV_ds_*/*kT*)) can be ignored. Thus, the effect of the drain voltage on the subthreshold current can be negligible. Although dropped, since the effective drain bias is greater than a few *q*/*kt*, the degradation could be observed in their transfer characteristics.
(3)$$I_{sub}=Wt_{ch}D_{n}\frac{dn}{dx}$$
(4)$$I_{sub}=\frac{W}{L}{t_{ch}D}_{n}\left\{n\left(x=L\right)-n\left(x=0\right)\right\}$$
(5)$$I_{sub}=\frac{W}{L}D_{n}T_{ch}n_{i}\exp\left(\frac{q\phi_{s}}{kT}\right)\left[1-\exp\left(-\frac{qV_{ds}}{kT}\right)\right]\;\approx \frac{W}{L}D_{n}T_{ch}n_{i}\exp\left(\frac{q\phi_{s}}{kT}\right)\;\;\;\;\;\;\;\;\;\;\;if\;V_{ds}\;\geq \;\frac{q}{kT}$$
where *D_n_* is the diffusion coefficient of electrons, *T_ch_* is the channel thickness, *n_i_* is the electron density, and *ϕ_s_* is the surface potential.

Another possible reason could be the disordered electronic features, such as the non-uniform interface trap states. As the gate bias increases, if the disorderedness of ZnO films increases, charge transports in the films would be seriously hindered, resulting in the degradation of the subthreshold swing. Hence, to unveil the effects of the disordered electronic features of the ZnO films on the degradation, we tried to modify the subthreshold current of Equation (5). Please note that, unless we use van der Waals electrodes such as graphene, capacitance–voltage (*C–V*)-based methods would be challenging to use for the extraction of the interfacial trap capacitance due to the doping effects from the top electrodes such as aluminum [22]. First, we tried to include the non-uniform and gate-dependent interface trap capacitance (*C_it_*) factor in the relation since the charges of the disordered semiconductors mainly transport via the localized trap states [25]. As depicted in Figure 3b and Equation (6), the surface potential (*ϕ_s_*) can be expressed in terms of the gate-bias-dependent interface trap capacitance of *C_it_*(*V_gs_*). By considering that the interface trap capacitance is far greater than the dielectric capacitance (*C_i_*) and that the semiconductor capacitance (*C_s_*) is negligible [7], the surface potential (*ϕ_s_*) can be given by Equation (7). Next, we tried to modify the diffusion coefficient of electrons (*D_n_*) for disordered semiconductors. For disorderless semiconductors, the diffusion coefficient of electrons can be expressed as *D_n_*/*μ_n_* = *kT*/*q* [14], whereas, for disordered semiconductors, it can be expressed as *D_n_*/*μ_n_* = *n*/*q*/(∂*n*/∂*E_F_*), where *E_F_* is the quasi-Fermi-level [34,35]. Thus, *D_n_* would also be a gate-bias-dependent factor for disordered semiconductors, which can be termed as *D_n_*(*V_gs_*). As a result, the subthreshold current for disordered semiconductors can be generally expressed as in Equation (8). In this study, for simplicity, we assume the factors can be expressed as gate-bias-dependent power laws as in Equation (9), of which the subthreshold current relation can be given by Equation (10), where *V_fb_* is the flatband voltage of the ZnO TFTs. In addition, the logarithmic derivation with the gate bias (*V_gs_*) of *dlnI_sub_*/*dV_gs_* can be derived as Equation (11), which can be useful for the extraction of the parameters such as the exponents and the flat voltage.
(6)$$C_{i}{(V}_{gs}-\phi_{s})=(C_{it}+C_{s})\phi_{s}$$
(7)$$\phi_{s}=\left(\frac{C_{i}}{C_{i}+C_{it}\left(V_{gs}\right)+C_{s}}\right)V_{gs}\;\approx \left(\frac{C_{i}}{C_{i}+C_{it}\left(V_{gs}\right)}\right)V_{gs}\;\approx \frac{C_{i}}{C_{it}\left(V_{gs}\right)}V_{gs}\;\;\;\;\;where\;C_{it}\gg C_{i},\;C_{s}$$
(8)$$I_{sub}\approx \frac{W}{L}D_{n}\left(V_{gs}\right)T_{ch}n_{i}\exp\left(\frac{q}{kT}\frac{C_{i}}{C_{it}\left(V_{gs}\right)}V_{gs}\right)$$
(9)$$D_{n}\left(V_{gs}\right)=\frac{kT}{q}\mu_{0}{\left(V_{gs}-V_{fb}\right)}^{\alpha },{\;C}_{it}\left(V_{gs}\right)=C_{it,0}{\left(V_{gs}-V_{fb}\right)}^{\beta }$$
(10)$$I_{sub}=\frac{W}{L}\frac{kT}{q}t_{ch}n_{i}\mu_{0}{\left(V_{gs}-V_{fb}\right)}^{\alpha }\exp\left(\frac{q}{kT}\frac{C_{i}}{C_{it,0}}{\left(V_{gs}-V_{fb}\right)}^{-\beta }V_{gs}\right)$$
(11)$$\frac{d\ln I_{Diff}}{dV_{gs}}=\frac{\alpha }{V_{gs}-V_{fb}}+\frac{q}{kT}\frac{C_{i}}{C_{it,0}}{\left(V_{gs}-V_{fb}\right)}^{-\beta }+\frac{q\left(V_{gs}\right)}{kT}\frac{C_{i}}{C_{it,0}}{(-\beta )\left(V_{gs}-V_{fb}\right)}^{-\beta -1}$$

Figure 3c presents the extraction plot of *dlnI_sub_*/*dV_gs_* for ZnO TFTs. From the nonlinear rational fitting of the plot, the coefficients of *α* and *β*, the flat voltage (*V_fb_*), and the coefficient of interface trap capacitance (*C_it_*_,0_) are estimated to be 0.92, 0.95, 6.9 V, and 0.5 μFcm^−2^, respectively. Using these parameters, as in the orange line of Figure 3c, *C_it_* is estimated to be 1.2 μFcm^−2^ at 10 V and 2.5 μFcm^−2^ at 15 V. Please note that the exponents of *α* can be determined as 0.92 by the proposed method [24], and the flat voltage (*V_fb_*) can be the turn-on voltage [36]. Then, to validate that the observed parameters are reliable, we tried to extract the interface trap capacitance using another current–voltage-based analyzing method of the *g_m_*/*I_ds_* method [27], of which the interface trap capacitance can be estimated as *C_it_*(*ϕ_s_*) = (*q*/*kT I_sub_*/*g_m_* − 1)*C_i_*. Although the method is developed for FinFETs, using the method, as depicted in the red line of Figure 3c, the interface trap capacitance was successfully extracted. Notably, the deduced trap capacitances by both methods were consistent with each other and are in order. Moreover, to further ensure the degradation of the subthreshold swing is attributed to the disorderedness of ZnO films, which are the gate-dependent diffusion coefficient of electrons and interface trap capacitance, we tried to count back the subthreshold swing using the deduced parameters from the *dlnI_sub_*/*dV_gs_* method. Remarkably, the estimated swings turned out to be quite consistent with the observed swing in Figure 3d. Hence, we strongly believe that the disorderedness of ZnO films results in the degradation, and the deduced disorderedness factors are reliable. In addition, we tried to figure out which conditions were for the steep subthreshold swings. In our calculations in Figure 3e, if the diffusion coefficient of electrons is constant, but the gate-dependent interface trap capacitance changes, as the gate bias increases, the subthreshold swing seriously degrades (green line in Figure 3e). Similarly, if the interface trap capacitance is constant at as high as 5 μFcm^−2^, but the gate-dependent diffusion coefficient of electrons changes, it also degrades (violet line in Figure 3e). However, if the interface trap capacitance is as low as 0.5 μFcm^−2^, it exhibits the nearly constant value of 1.7 V/dec (red line in Figure 3e), and if it is as extremely low as 5 nFcm^−2^, it can be minimized as 0.1 V/dec (orange line in Figure 3e). Thus, we can interpret the effects of the disorderedness of ZnO films on the subthreshold conduction as follows. As described in Equation (10), if the gate-dependent interface traps seriously increase as the gate bias increases, the injected charges from the source electrode would be trapped, resulting in a decrease in effective charge densities and the degradation of the subthreshold swing. In addition, if the injected charges transport faster (the diffusion coefficient of electrons increases as the gate bias increases), the subthreshold swing will enhance. Additionally, we posit that the interface traps result in the gate-dependent diffusion coefficient of electrons. If the interface traps exhibit large densities, the diffusion coefficient of electrons would be small due to the trapping. Hence, for the steep-subthreshold-swing TFTs, we speculate that the gate-dependent interface traps should be minimized. In addition, extremely high-capacitive dielectrics such as self-assembled nanodielectrics or ionic-liquid dielectrics can be alternative routes for mitigating the interface trap effects on the subthreshold swing [37,38]. Hence, for ensuring the high-capacitive dielectric effects on the subthreshold conduction behaviors of the ZnO TFTs, we fabricated ZnO TFTs using a 10 nm thin Al_2_O_3_ dielectric layer ($C_{i}=$521 nF cm^–2^, Appendix A, for the thin dielectric capacitance). As in Figure 4a, low-voltage ZnO TFTs were successfully fabricated: the field-effect mobility (*μ*) is extracted to be 3.5 cm^2^ V^−1^ s^−1^ at *V_gs_* = 2.5 V, and a threshold voltage (*V_th_*) is deduced to be 0.7 V along with a high on/off ratio (>10^5^). The output characteristics (*I_d_*_s_ vs.*V_ds_*) of the low-voltage ZnO TFTs are presented in Figure 4b. The channel width and length of the ZnO TFTs are 300 and 60 μm, respectively. As in Figure 4c, although the degradation of the subthreshold swing is still observed, which is minimally extracted to 0.1 V/dec at *V_gs_* = 0.4 V, but increasing to 0.2 V/dec at *V_gs_* = 0.7, the subthreshold swing was significantly lower than that of the ZnO TFTs on SiO_2_. More remarkably, the gate-dependent interface trap capacitance of the ZnO TFTs using the *dlnI_sub_*/*dV_gs_* and *g_m_*/*I_ds_* methods was estimated to be 0.5 μFcm^−2^ at 0.4 V and 1.9 μFcm^−2^ at 0.7 V in Figure 4d, which is comparable to that of the ZnO TFTs on SiO_2_. The coefficients of *α* and *β*, the flat voltage (*V_fb_*), and the coefficient of interface trap capacitance (*C_it_*,*_0_*) are estimated to be 1.83, 1.02, 0.3 V, and 4.6 μFcm^−2^, respectively. Thus, although the gate-dependent interface capacitance is disordered, a relatively steep subthreshold swing can be achieved by using high capacitive dielectrics.

Furthermore, to gain insights into the subthreshold conduction of the disordered ZnO TFTs, we investigated and carefully analyzed the temperature-dependent current–voltage characteristics of the ZnO TFTs from 180 to 300 K. Figure 5a shows the temperature-dependent transfer characteristics of the ZnO TFTs. By assuming that the gate-dependent activation energy (*E_a_*) is closely related to the energetic difference between the Fermi level and conductive states, the gate-dependent activation energy can be extracted using the Meyer–Neldel rule of *I*(*V_gs_*) = *I*_0_ exp(−*E_a_*/*kT*), as in the inset of Figure 5a [22,39]. From the gate-dependent activation energy, the areal density of states, DOS, *g*(*E*) can be deduced using the relation of *g*(*E*) = *qC_i_* (*dE_a_*/*dV_gs_*)^−1^, as in Figure 5b [40,41]. At the low-gate-bias regime (*V_gs_* < 15 V), as the gate bias increased, it increased rapidly to ~10^13^ states eV^−1^cm^2^, whereas, at the high-gate-bias regime (*V_gs_* > 15 V), it slowly increased to ~10^14^ states eV^−1^cm^2^. As reported elsewhere [39], these abrupt changes in the trap DOS of the ZnO TFTs are attributed to the conduction route change from diffusion to drift, of which the *V_gs_* of 15 V can be defined as the threshold voltage of the disordered ZnO TFTs. Notably, when the interface trap density (*D_it_*) is deduced using the relation of *C_it_* = *q*^2^*D_it_* [23] and compared with the DOS, as in orange in Figure 5b, the crossover point of the 15 V can be observed, which is strongly regarded as the threshold voltage. Thus, it can be interpreted as follows: at the low-gate-bias regime (*V_gs_* < 15 V), due to the relatively high density of the interface traps, injected charges from the source electrode are severely trapped, resulting in the low density of the DOS. As a result, the current would be driven by the concentration gradient (diffusion current). However, at the high-gate-bias regime (*V_gs_* < 15 V), due to the increased injected charges, the interface trap states are almost filled with the charges, leading to the relatively high density of the DOS, thereby leading to the electric-field-driven currents (drift current). The subthreshold conduction analyses employing other disordered semiconductors such as p-type or ambipolar semiconductors would be required for proving the method is applicable in more general. In addition, we currently use Al as electrodes for ZnO TFTs. Although potential drops were observed, the effective drain voltage was larger than a few *q*/*kt*, and the TFTs exhibit ohmic characteristics, which enables us to neglect the drain bias effects on the subthreshold conduction. However, if we use Au or Pt as electrodes, due to the high work function of the electrodes, the drain bias effects should be considered. Hence, device analyses using high-Schottky contact devices would be essential for an in-depth understanding of the subthreshold behaviors of the disordered semiconductor TFTs. Nevertheless, based on the results, we believe that the proposed current–voltage-based analyzing method is reliable and applicable for extracting the disordered electronic features of the disordered semiconductor TFTs.

## 4. Conclusions

In this study, we investigated the subthreshold characteristics of atomically deposited ZnO TFTs. The ZnO TFTs exhibit conventional n-type enhancement-mode transfer characteristics with an electronic field-effect mobility of 3.1 cm^2^ V^−1^ s^−1^. However, the degradation of the subthreshold swing appears, which is 1.8 V/dec at *V_gs_* = 9 V, but increases to 8.1 V/dec at *V_gs_* = 15 V. By considering that the degradation is attributed to the gate-dependent disorderedness of the ZnO films, the subthreshold current relation was derived by introducing the gate-dependent interface trap capacitance and the diffusion coefficient of electrons (the *dlnI_sub_*/*dV_gs_* method). Remarkably, using the method, the gate-dependent disorderedness factors were successfully extracted and consistent with those obtained by the *g_m_*/*I_ds_* method. Furthermore, the threshold voltage of ZnO TFTs (~15 V) was identified when the interface trap density was compared with the trap density of states by low-temperature current–voltage analyses. We thus conclude that the degradation of the subthreshold characteristics of the ZnO TFTs is attributed to the gate-dependent disorderedness factors, and our proposed approaches would offer practical and reliable tools for understanding the subthreshold conduction of disordered semiconductors.

## Figures and Tables

**Figure 1 micromachines-14-01596-f001:**
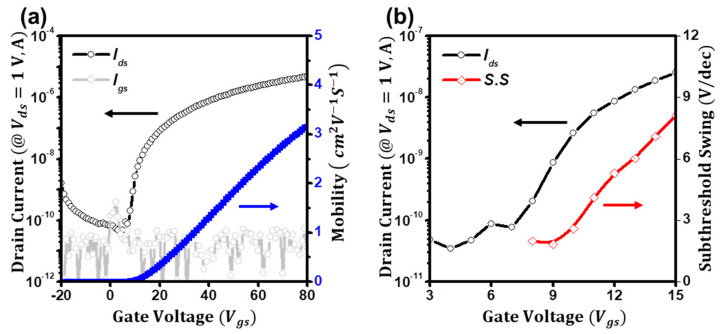
(**a**) Transfer characteristics (*I_ds_* vs. *V_gs_*) of the ZnO TFTs. Blue line: Field−effect mobility. (**b**) Subthreshold swing of the ZnO TFTs.

**Figure 2 micromachines-14-01596-f002:**
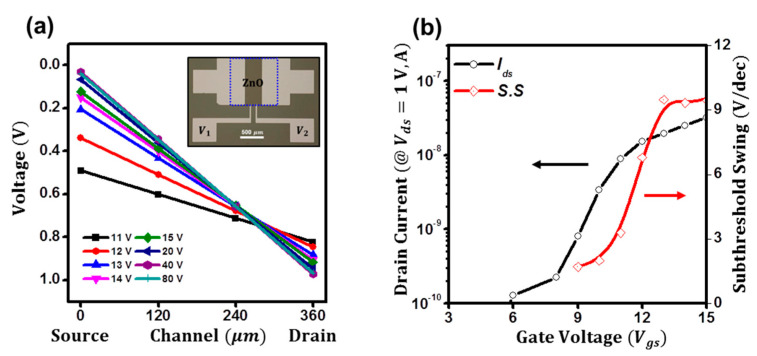
(**a**) The potential distributions of the ZnO TFTs. (**b**) Intrinsic subthreshold swing of the ZnO TFTs.

**Figure 3 micromachines-14-01596-f003:**
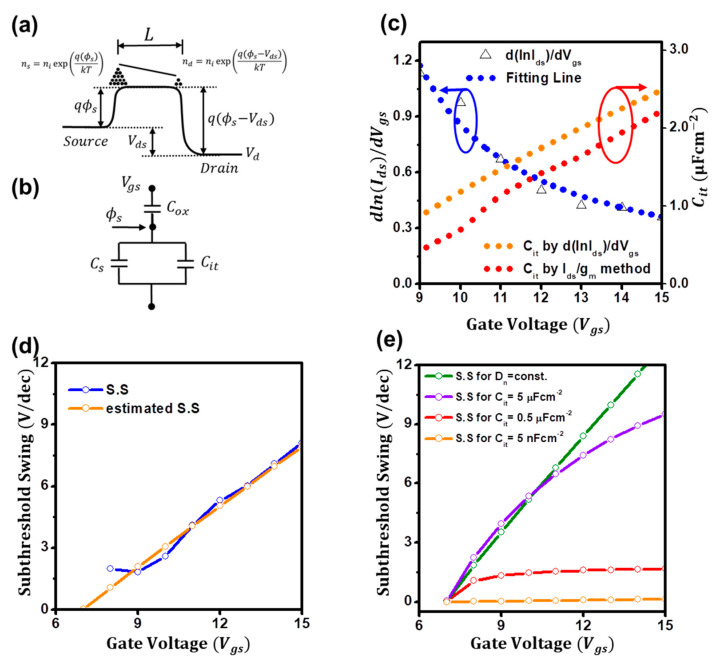
(**a**) Schematic illustration of the diffusion current of the ZnO TFTs. (**b**) Equivalent circuits with an interface trap capacitance of the ZnO TFTs. (**c**) Extraction plots for the interface trap capacitance. (**d**) Estimated subthreshold swings using the extracted parameters from the *dlnI_sub_*/*dV_gs_* method. (**e**) Estimated subthreshold swings for *C_it_* and *D_n_* are constant.

**Figure 4 micromachines-14-01596-f004:**
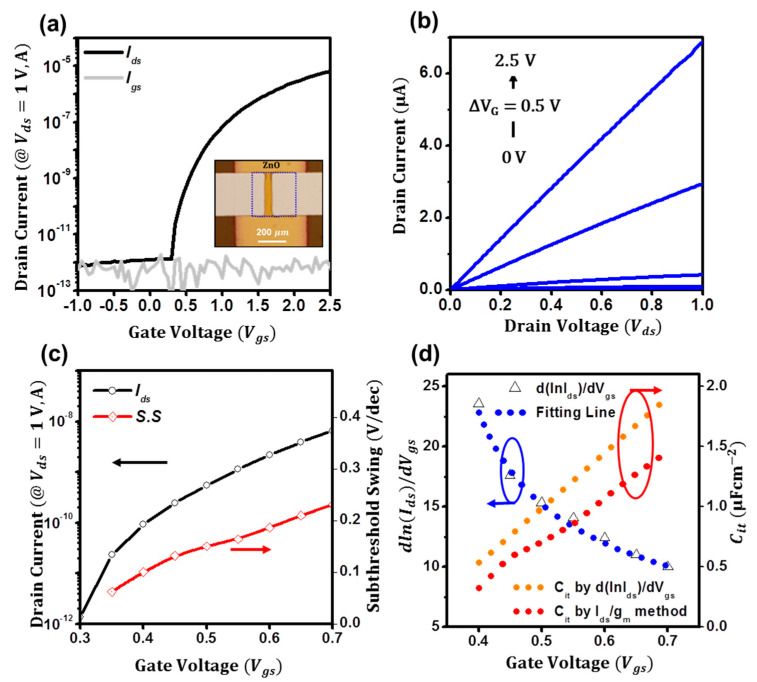
(**a**,**b**) Transfer and output characteristics of the low−voltage ZnO TFTs. (**c**) Subthreshold swing of the low−voltage ZnO TFTs. (**d**) Extraction plots for the interface trap capacitance.

**Figure 5 micromachines-14-01596-f005:**
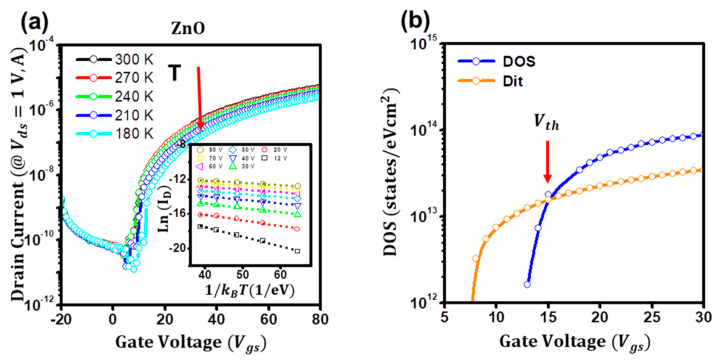
(**a**) Temperature dependence of the transfer curves of the ZnO TFTs. Inset: Meyer−Neldel plots of *Ln*(*I_D_*) as a function of 1/*T* of the ZnO TFTs. (**b**) Extracted trap density of states (blue) and the interface trap states (orange) of the ZnO TFTs.

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
