# Peer review of "Subthreshold Conduction of Disordered ZnO-Based Thin-Film Transistors"

_micromachines, 2023, doi:10.3390/mi14081596_

Round 1

Reviewer 1 Report

The authors of manuscript micromachines-2560007, entitled "Subthreshold conduction of disordered ZnO-based thin-film transistors", presents the disorderedness effects on the subthreshold characteristics of the 7 atomically deposited ZnO thin-film transistors (TFTs).

In this manuscript, the ZnO TFTs exhibit conventional n-type enhancement-mode transfer characteristics. However, the degradation of the subthreshold swing appears. By considering that the degradation is attributed to the gate-dependent disorderedness of the ZnO films, the subthreshold current relation was derived by introducing the gate-dependent interface trap capacitance and the diffusion coefficient of electrons. Using the method, the gate-dependent disorderedness factors were successfully extracted. Furthermore, the threshold voltage of ZnO TFTs (~15 V) was identified when the interface trap density was compared with the trap density of states by low-temperature current–voltage analyses. Author thus concludes that the degradation of the subthreshold characteristics of the ZnO TFTs is attributed to the gate-dependent disorderedness factors. However, some issues should be concerned:

1.      Author posits that the disorderedness of the ZnO films would result in the degradation; hence he derived the subthreshold current–voltage relation-ship by considering the disorderedness factors. I suggest that author clarify the physical mechanism how the disorderedness of the ZnO films would result in the degradation.

2.      Fig 2a, it is not clear how to extract the potential distributions of the ZnO TFTs.

Author Response

Summary of Revisions & Responses to Reviewers' Comments

We have revised the manuscript to address the reviewers’ comments. Our response to each comment is given below in detail. In the following, the reviewers’ comments are in black fonts and our replies are in blue fonts.

REVIEWER REPORT(S):
Referee: 1

The authors of manuscript micromachines-2560007, entitled "Subthreshold conduction of disordered ZnO-based thin-film transistors", presents the disorderedness effects on the subthreshold characteristics of the atomically deposited ZnO thin-film transistors (TFTs).

In this manuscript, the ZnO TFTs exhibit conventional n-type enhancement-mode transfer characteristics. However, the degradation of the subthreshold swing appears. By considering that the degradation is attributed to the gate-dependent disorderedness of the ZnO films, the subthreshold current relation was derived by introducing the gate-dependent interface trap capacitance and the diffusion coefficient of electrons. Using the method, the gate-dependent disorderedness factors were successfully extracted. Furthermore, the threshold voltage of ZnO TFTs (~15 V) was identified when the interface trap density was compared with the trap density of states by low-temperature current–voltage analyses. Author thus concludes that the degradation of the subthreshold characteristics of the ZnO TFTs is attributed to the gate-dependent disorderedness factors. However, some issues should be concerned:

  1. Author posits that the disorderedness of the ZnO films would result in the degradation; hence he derived the subthreshold current–voltage relation-ship by considering the disorderedness factors. I suggest that author clarify the physical mechanism how the disorderedness of the ZnO films would result in the degradation.

Response) Thank you for your valuable comments. As described in Equation (10), if the gate-dependent interface traps seriously increase as the gate bias increases, the injected charges from the source electrode would be trapped, resulting in a decrease in effective charge densities and the degradation of the subthreshold swing. In addition, if the injected charges transport faster (the diffusion coefficient of electrons increases as the gate bias increases), the subthreshold swing would enhance. Besides, we posit that the interface traps result in the gate-dependent diffusion coefficient of electrons. If the interface traps exhibit large densities, the diffusion coefficient of electrons would be small due to the trapping. Hence, for the steep subthreshold swing TFTs, we speculate that the gate-dependent interface traps should be minimized. To help readers understand this issue, we add the following sentences in the manuscripts. Thank you very much.

Thus, we can interpret the effects of the disorderedness of ZnO films to the subthreshold conduction as followings. As described in Equation (10), if the gate-dependent interface traps seriously increase as the gate bias increases, the injected charges from the source electrode would be trapped, resulting in a decrease in effective charge densities and the degradation of the subthreshold swing. In addition, if the injected charges transport faster (the diffusion coefficient of electrons increases as the gate bias increases), the subthreshold swing would enhance. Besides, we posit that the interface traps result in the gate-dependent diffusion coefficient of electrons. If the interface traps exhibit large densities, the diffusion coefficient of electrons would be small due to the trapping. Hence, for the steep subthreshold swing TFTs, we speculate that the gate-dependent interface traps should be minimized.

  1. Fig 2a, it is not clear how to extract the potential distributions of the ZnO TFTs.

Response) By considering the potential in the channel is linearly distributed, the potential drops at the source and drain electrodes (ΔVS, ΔVD) can be deduced by measuring potentials with the voltage probes (V1 and V2), which are given by Equations below.

where V1 and V2 are measured voltages with potential probes, and L1, L2, and L are the distance from the source electrode to the first, second, and drain electrodes, respectively. Thus, we extract the potential distributions of the ZnO TFTs using the potential drops and measured potentials. In this study, the voltage probing electrodes (V1 and V2) were placed at 120 and 240 μm in the channel. To help readers understand this issue, we add the following sentences in the manuscripts.

“For extracting the potential distributions of the ZnO TFTs, we use the four-probe method. By considering the potential in the channel is linearly distributed, the potential drops at the source and drain electrodes (ΔVS, ΔVD) can be deduced by measuring potentials with the voltage probes (V1 and V2), which are given by Equation (2).

(2)

where V1 and V2 are measured voltages with potential probes, and L1, L2, and L are the distance from the source electrode to the first, second, and drain electrodes, respectively. Thus, we extract the potential distributions of the ZnO TFTs using the potential drops and measured potentials. In this study, the voltage probing electrodes (V1 and V2) were placed at 120 and 240 μm in the channel.”

“Please see the material and methods section for more information on the four-probe method”

Reviewer 2 Report

The manuscript demonstrated the ZnO-based TFTs with special subthreshold conduction behavior. There are some contents that should be added to the manuscript before the acceptance of it.

1. What is the structure of disordered ZnO? The structure information should be offered, such as TEM images and SAED patterns.

2. In Figure 1, the leakage current of the TFT (Igs) is ~0.1 nA, which is too large to confirm the excellent performance of the ZnO TFT.

3. The work functions of electrodes and the Al2O3 layer may affect the subthreshold conduction behavior. TFTs with different electrodes and dielectric layers should be compared with the ZnO TFT.

Author Response

Summary of Revisions & Responses to Reviewers' Comments

We have revised the manuscript to address the reviewers’ comments. Our response to each comment is given below in detail. In the following, the reviewers’ comments are in black fonts and our replies are in blue fonts. Please see the attached file. 
